

# Crystalline Weyl semimetal phase
# in quantum spin Hall systems under magnetic fields

**Fernando Dominguez⋆, Benedikt Scharf and Ewelina M. Hankiewicz**

Institute for Theoretical Physics and Astrophysics and Würzburg-Dresden Cluster of Excellence ct.qmat, University of Würzburg, Am Hubland, 97074 Würzburg, Germany

⋆ fernando.dominguez@physik.uni-wuerzburg.de

## Abstract

We investigate an unconventional topological phase transition that occurs in quantum spin Hall (QSH) systems when applying an external in-plane magnetic field. We show that this transition between QSH and trivial insulator phases is separated by a stable topological gapless phase, which is protected by the combination of particle-hole and reflection symmetries, and thus, we dub it as crystalline Weyl semimetal. We explore the stability of this new phase when particle-hole symmetry breaking terms are present. Especially, we predict a robust unconventional topological phase transition to be visible for materials described by the Kane and Mele model even if particle-hole symmetry is significantly broken.

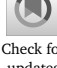
# 1  Introduction

Weyl fermions can be realized as quasiparticle excitations of certain materials, the Weyl se-
mimetals (WSM). They adopt their name from the analogy with the Weyl Hamiltonian, i.e. the
Hamiltonian of a (3+1)D massless relativistic particle. Recently, this novel phase has attracted
great interest due to its realization in the laboratory [1–15] and due to novel physics associated
with its topologically non-trivial character: Weyl nodes (WN) appear always in pairs with
opposite chirality [16] and a Fermi arc connects them in momentum space [9, 17–24, 24–35].

Weyl semimetals originate from stable accidental crossings between a pair of bands. In
contrast to symmetry protected crossing points, WN are stable to perturbations, which can
only shift their position in reciprocal space but not their presence. These gapless points carry
a quantized topological invariant, which guarantees their stability. The presence of these sta-
ble points can be formally understood by studying the codimension of the Hamiltonian, i.e. the
number of tunable parameters to achieve degeneracy. This number is sensitive to the symme-
tries and dimensionality of the Hamiltonian. For example, assuming a lack of time-reversal
symmetry (TRS) and/or inversion symmetry (IS), we can write a generic two-band Hamilto-
nian

$$H(\mathbf{k}) = f(\mathbf{k})\sigma_0 + \sum_{i=x,y,z} f_i(\mathbf{k})\sigma_i, \tag{1}$$

with eigenenergies $E_\pm = f(\mathbf{k}) \pm \sqrt{\sum_{i=x,y,z} f_i^2(\mathbf{k})}$ and the Pauli matrices $\sigma_i$ spanning the space
of the two bands. Here, the band crossing occurs when $f_x = f_y = f_z = 0$ are fulfilled.[1] In
three dimensions (3D), there are four independent parameters available $(k_x, k_y, k_z, m)$ (three
momenta and the mass), and therefore, it is natural to expect a gap closing for a finite range of
$m$. An example of this is 3D non-centrosymetric topological insulators, where IS is broken [36].
In contrast, in 2D it is only possible to find at most one solution by fine tuning $(k_x, k_y, m)$,
instead of a range of points. However, when more symmetries are involved, a crystalline WSM
phase can also emerge in 2D systems at high-symmetry points or lines [37]. Indeed, it has been
recently shown that QSH systems with twofold rotational symmetry exhibit a WSM phase when
an inversion symmetry breaking contribution is added [38]. Here, the combination of TRS
and rotational symmetry relaxes the conditions to find stable crossing points in a 2D **k**-space.
This crystalline WSM phase mediates the topological phase transition between the topological
and trivial insulator phases for a finite range of points in the parameter space. In contrast
to conventional topological phase transitions, where the gap closes at one single point in the
parameter space, these transitions are called unconventional. Similar phase transitions can
occur in bilayer setups of Rashba 2D electron gases subject to superconductivity, where a Weyl

---

[1]Note that in the presence of both TRS and IS, every **k**-point is doubly degenerated, yielding a double Hilbert
space and two extra conditions have to be fulfilled.

phase separates trivial from second order topological superconducting phases in $\pi$-junction Rashba layers [39, 40]. Likewise, Floquet Weyl phases can be situated between trivial and topological Floquet phases [41].

In this paper, we study an unconventional topological phase transition of a QSH system driven by an external in-plane magnetic field. This topological phase transition corresponds to the TRS broken counterpart of the mentioned IS breaking topological phase transition [38]. However, in contrast to that case, the difficulty to induce such topological phase transitions comes from the fact that a magnetic field removes the protection from the QSH phase giving rise to the opening of a gap. We overcome this difficulty taking advantage of a topological crystalline protection against magnetic fields present in QSH systems with particle-hole symmetry and a commuting reflection symmetry [42, 43]. Recently, we found an example of this phenomenon in bismuthene on SiC [42], a hexagonal lattice QSH system that offers exciting prospects for room-temperature applications [44, 45]. There, the combination of particle-hole and reflection symmetries prevents the mixing between counter-propagating edge states preserved only along the armchair boundary [42]. In this paper, we demonstrate that the topological crystalline protection and formation of crystalline WSM phase occurs in all hexagonal lattice QSH systems described by the Kane-Mele (KM) Hamiltonian [46–57] and by the Bernevig-Hughes-Zhang (BHZ) model, although in practice it is difficult to realize particle-hole symmetry (PHS) in semiconductors such as HgTe/CdTe [58–63] or InAs/GaSb [64] quantum wells. Furthermore, we explore the stability of the crystalline WSM phase when PHS is broken, and predict the robustness of this phase in systems described by KM model.

The structure of the paper is as follows: In Sec. 2, we introduce the KM and BHZ models. Then, in Sec. 3 we classify and calculate the topological invariants of the aforementioned Hamiltonians, only considering the particle-hole symmetric terms together with spatial symmetries. In Sec. 4, we study the gap opened by the in-plane magnetic field when terms that break PHS are present. Additionally, in Sec. 5 we study the effect of disorder on the two terminal conductance in the unconventional topological phase transition. Finally in Sec. 6, we explain a potential realization on topoelectrical circuits.

## 2 Kane-Mele and BHZ models: Hamiltonian and symmetries

First, we give a brief overview of the two paradigmatic QSH models studied here, the KM and BHZ models. For both models, we use a tight-binding (TB) description, set up on a hexagonal (KM) or square (BHZ) lattice. In addition to the Hamiltonians, we also provide a brief survey of the symmetries and high-symmetry points involved in each model.

### 2.1 Kane-Mele model

#### 2.1.1 Particle-hole symmetric terms

The KM model was the first theoretical example of a QSH insulator, synonymously also referred to as two-dimensional (2D) topological insulator [65, 66]. It describes a QSH system on a hexagonal lattice, with a single orbital per atom (and spin). Although originally proposed for graphene [65, 66], it has subsequently been used to describe other hexagonal QSH systems, such as silicene [46–49], germanene [46, 50], stanene [51, 52], or jacutingaite [55]. In reciprocal space, the bulk Hamiltonian of the KM model can be written as

$$H_0^{\text{KM}} = f_x \sigma_x + f_y \sigma_y + f_z \sigma_z s_z \,, \tag{2}$$

where $f_x = -t\left[1 + 2\cos\left(k_x a/2\right)\cos\left(\sqrt{3}k_y a/2\right)\right]$, $f_y = 2t\cos\left(k_x a/2\right)\sin\left(\sqrt{3}k_y a/2\right)$, as well as $f_z = \left(2\lambda_{\text{SOC}}/3\sqrt{3}\right)\left[\sin\left(k_x a\right) - 2\sin\left(k_x a/2\right)\cos\left(\sqrt{3}k_y a/2\right)\right]$. Here, the direct Bravais lat-

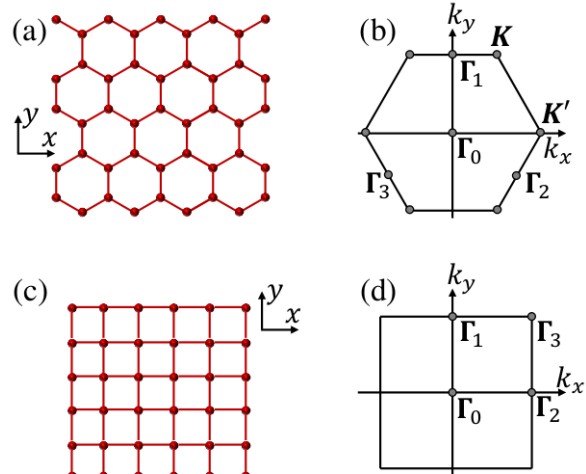

Figure 1: (Color online) (a) Direct and (b) reciprocal lattice for the honeycomb-based systems described by the KM Hamiltonian. (c) Direct and (d) reciprocal lattice for the BHZ Hamiltonian discretized on a square lattice. Here, the time-reversal invariant momenta $\Gamma_{0-3}$ of the hexagonal and square lattices are also indicated.

tice vectors have been chosen as $\mathbf{a}_1 = a\mathbf{e}_x$ and $\mathbf{a}_2 = -(a/2)\mathbf{e}_x + (\sqrt{3}a/2)\mathbf{e}_y$ with the lattice constant $a$, such that the $x$ and $y$ directions denote the zigzag (ZZ) and armchair (AC) directions, respectively [see Fig. 1(a) for the honeycomb lattice and the directions]. The nearest-neighbor hopping amplitude is given by $t$ and the intrinsic spin-orbit coupling parameter by $\lambda_{\text{SOC}}$. Moreover, we have introduced the Pauli matrices $\sigma_{x,y,z}$ and $s_{x,y,z}$ to describe the sublattice and spin degrees of freedom, respectively.

Apart from the $\mathbf{K}$ and $\mathbf{K}'$ points, the Hamiltonian (2) possesses other high-symmetry points, namely the time-reversal invariant momenta, which fulfill $H(k_x, k_y) = H(-k_x, -k_y)$ and are given by

$$\Gamma_0 = (0,0)\,, \quad \Gamma_1 = \frac{2\pi}{a}\left(0, \frac{1}{\sqrt{3}}\right), \quad \Gamma_2 = \frac{2\pi}{a}\left(\frac{1}{2}, -\frac{1}{2\sqrt{3}}\right), \quad \Gamma_3 = \frac{2\pi}{a}\left(-\frac{1}{2}, -\frac{1}{2\sqrt{3}}\right).$$

Note that we have introduced here a generic notation $\Gamma_i$, to make it coincident with the one used in the BHZ model. In addition, we introduce the reflection-invariant momenta, which have the property $H(k_x, k_y) = H(k_x, -k_y)$ and are given by $k_y = 0$ and $k_y = 2\pi/(\sqrt{3}a)$.

An applied in-plane magnetic field $\mathbf{H}$ gives rise to a Zeeman term

$$H_Z = B_{\|}\left[\cos(\theta)s_x + \sin(\theta)s_y\right] \equiv B_{\|}s_\theta\,, \tag{3}$$

with the Zeeman energy $B_{\|}$ and the angle $\theta$ between the $x$ direction and the orientation of the magnetic field. The Zeeman energy is given by $B_{\|} = g\mu_B|\mathbf{H}|/2$, where $\mu_B$ is the Bohr magneton, $|\mathbf{H}|$ the magnitude of the in-plane magnetic field, and $g$ the material-dependent g-factor. If $g = 2$, $B_{\|} = \mu_B|\mathbf{H}|$.

In the presence of $H_Z$, the reflection and PHS operators are given by

$$\mathcal{C} = \sigma_z s_z \exp(-i\theta s_z)\mathcal{K}\,, \tag{4}$$

$$\mathcal{R}(x) = \sigma_0 s_\theta\,, \tag{5}$$

$$\mathcal{R}(y) = \sigma_x s_\theta\,, \tag{6}$$

where $\mathcal{K}$ denotes complex conjugation. The bulk Hamiltonian given by Eq. (2) exhibits reflection symmetries, i.e.,

$$\mathcal{R}(i)H(\overline{\mathbf{k}})\mathcal{R}^{-1}(i) = H(\mathbf{k}), \tag{7}$$

where $i = x, y$, $\mathbf{k} = (\mathbf{k_x}, \mathbf{k_y})$ and $\overline{\mathbf{k}}$ is equal to $\mathbf{k}$ except for its $i$th-component, which is reflected ($k_i \rightarrow -k_i$). Note that $\mathcal{R}(x)$ and $\mathcal{R}(y)$ change with the direction of the in-plane magnetic field angle $\theta$.

### 2.1.2 Terms breaking particle-hole symmetry

In systems without superconductivity, PHS is usually broken when additional coupling terms, present in real materials, are accounted for. In honeycomb lattices, one of the most important examples for such terms constitutes Rashba spin-orbit coupling

$$
\begin{aligned}
H_{\mathrm{R}} =& \lambda_{\mathrm{R}}\sqrt{3}\sin\left(\frac{k_x a}{2}\right)\cos\left(\frac{\sqrt{3}k_y a}{2}\right)\sigma_x s_y - \lambda_{\mathrm{R}}\sqrt{3}\sin\left(\frac{k_x a}{2}\right)\sin\left(\frac{\sqrt{3}k_y a}{2}\right)\sigma_y s_y \\
&+ \lambda_{\mathrm{R}}\left[1 - \cos\left(\frac{k_x a}{2}\right)\cos\left(\frac{\sqrt{3}k_y a}{2}\right)\right]\sigma_y s_x - \lambda_{\mathrm{R}}\sin\left(\frac{\sqrt{3}k_y a}{2}\right)\cos\left(\frac{k_x a}{2}\right)\sigma_x s_x, \quad (8)
\end{aligned}
$$

where $\lambda_{\mathrm{R}}$ is the Rashba spin-orbit constant.

Another important term breaking PHS in honeycomb lattices is next-nearest-neighbor hopping, described by the Hamiltonian

$$H_{\mathrm{NNN}} = t'\sigma_0 s_0\left[\cos(k_x a) + 2\cos\left(\frac{k_x a}{2}\right)\cos\left(\frac{\sqrt{3}k_y a}{2}\right)\right], \tag{9}$$

where $t'$ is the next-nearest-neighbor hopping amplitude, which in graphene is only a fraction of the nearest-neighbor amplitude $t' \sim 0.1t$ [67]. While there are other terms breaking PHS, we have focused here on the terms that induce the most sizable gap openings between the edge states. In addition, we consider the impact of disorder on the two terminal conductance, see below.

## 2.2 BHZ model

### 2.2.1 Particle-hole symmetric terms

The BHZ model has first been proposed to describe the low-energy physics of HgTe/CdTe quantum-well structures, which in the inverted regime host QSH edge states [59–61, 63]. Not only does the BHZ model describe HgTe/CdTe quantum wells, the first experimentally demonstrated QSH insulator [59], but also other systems such as InAs/GaSb [64] quantum wells.

The continuum BHZ model can be discretized on a square lattice to obtain an effective tight-binding description with lattice constant $a$ [see Figs. 1(c) and (d)]. If only the terms preserving PHS are taken into account, the resulting Hamiltonian is given by

$$H_0^{\mathrm{BHZ}} = f_x\sigma_x s_z + f_y\sigma_y s_0 + f_z\sigma_z s_0, \tag{10}$$

where $f_x = \mathcal{A}\sin(k_x a)$, $f_y = -\mathcal{A}\sin(k_y a)$, and $f_z = \mathcal{M} - \mathcal{B}\cos(k_x a) - \mathcal{B}\cos(k_y a)$. Now, $\sigma_{x,y,z}$ are Pauli matrices that represent the electron-like and heavy hole-like states. As before $s_{x,y,z}$ are spin Pauli matrices. $\mathcal{A}$, $\mathcal{B}$, and $\mathcal{M}$ are material parameters depending on the quantum-well thickness $d$ (along the $z$-direction) and also on the finite lattice constant/step size $a$. The

system becomes topological for $\mathcal{M} < 2\mathcal{B}$, which in HgTe quantum wells occurs if $d$ exceeds a critical thickness.

For the square lattice, the time-reversal invariant momenta are given by

$$\Gamma_0 = (0,0)\,, \quad \Gamma_1 = \frac{\pi}{a}(0,1)\,, \quad \Gamma_2 = \frac{\pi}{a}(1,0)\,, \quad \Gamma_3 = \frac{\pi}{a}(1,1)\,,$$

while the reflection-invariant momenta with the property $H(k_x, k_y) = H(k_x, -k_y)$ are given by $k_y = 0$ and $k_y = \pi/a$.

In the presence of an external in-plane magnetic field $H$, which–if particle-hole symmetric–has the form of Eq. (3), the Hamiltonian preserves PHS as well as reflection symmetry. The PHS and reflection operators are described by

$$\mathcal{C} = i\sigma_x s_z \exp(i\theta s_z)\mathcal{K}\,, \tag{11}$$

$$\mathcal{R}(x) = \sigma_0 s_\theta\,, \tag{12}$$

$$\mathcal{R}(y) = \sigma_z s_\theta\,, \tag{13}$$

for the square lattice. Again, $\mathcal{K}$ denotes complex conjugation here.

### 2.2.2 Terms breaking particle-hole symmetry

In materials described by the BHZ model, there are typically sizable contributions that break PHS. The most important contribution of these is given by

$$H_\mathrm{D} = \left[\mathcal{E} - \mathcal{D}\cos(k_x a) - \mathcal{D}\cos(k_y a)\right]\sigma_0 s_0\,, \tag{14}$$

which describes the asymmetry between the effective masses of the electron- and hole-like bands. Again, $\mathcal{E}$, $\mathcal{D}$, and $\mathcal{M}$ are material parameters depending on the quantum-well thickness $d$ and the lattice constant $a$.

Moreover, additional spin-orbit contributions of the quantum-well structure are described by [61,68]

$$H_\mathrm{R} = \xi_e \frac{\sigma_0 + \sigma_z}{2}\left[\sin(k_y a)s_x - \sin(k_x a)s_y\right]\,, \tag{15}$$

where $\xi_e$ depends not only on the quantum-well width, but also on the lattice constant $a$. Finally, the in-plane magnetic field can give rise to a Zeeman term

$$H_g = B_{\|,e}\left[\cos\theta\frac{\sigma_0 + \sigma_z}{2}s_x + \sin\theta\frac{\sigma_0 + \sigma_z}{2}s_y\right] + B_{\|,h}\left[\cos\theta\frac{\sigma_0 - \sigma_z}{2}s_x + \sin\theta\frac{\sigma_0 - \sigma_z}{2}s_y\right] \tag{16}$$

that by itself breaks PHS for $B_{\|,e} \neq B_{\|,h}$. Here, an in-plane magnetic field $\mathbf{H}$ can give rise to different Zeeman energies $B_{\|,e} = g_e\mu_B|\mathbf{H}|/2$ and $B_{\|,h} = g_h\mu_B|\mathbf{H}|/2$ of the electron- and hole-like bands, respectively [69]. These different Zeeman energies are due to different in-plane g factors $g_e$ and $g_h$ in HgTe quantum wells. As before, $\theta$ describes the angle between the orientation of $\mathbf{H}$ and the $x$ axis. Like the other material parameters, the g factors depend on the thickness $d$ of the quantum well. While there are other terms breaking PHS, we have focused here on the terms that induce the most sizable gap openings between the edge states. Finally, we note that bulk inversion asymmetry, described by $H_{BIA} = \Delta_{BIA}\sigma_y s_y$ with the strength $\Delta_{BIA}$ [61], can play an important role in the BHZ model. We find, however, that $H_{BIA}$ does not break PHS and reflection symmetry in finite strips that are infinite in the $y$ direction. Hence, the edge states remain topologically protected and gapless in these strips even for $\Delta_{BIA} \neq 0$.

In the following paragraphs, we will calculate the spectra for AC/ZZ nanoribbons or finite strips in the BHZ model. To do so, we discretize the Hamiltonian given by Eqs. (2) and (10) in one of the directions and leave the other direction with the good quantum number $k$. Further details are given in App. A.

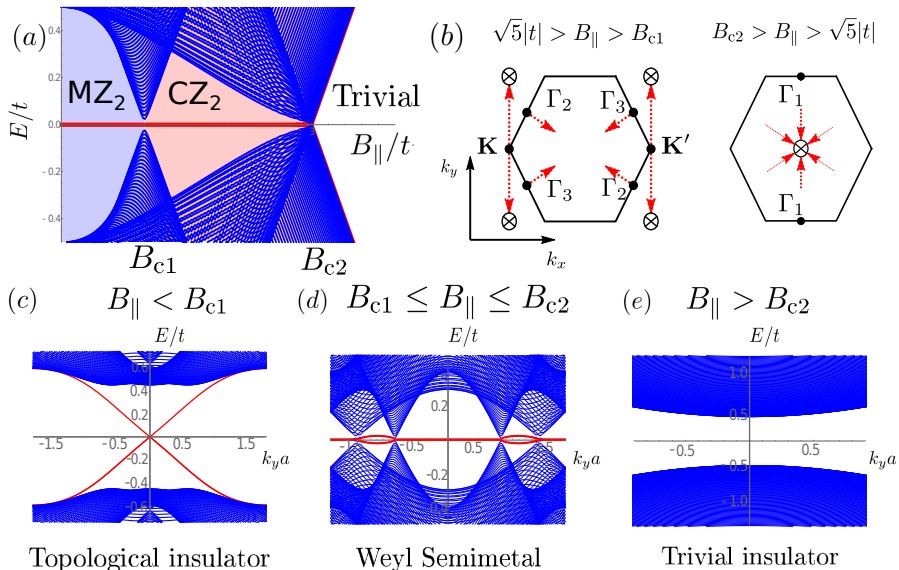

Figure 2: (Color online) In panel (a), we show the energy spectrum as a function of $B_\parallel$ for an AC nanoribbon at $k_y = 0$. Here, blue (red) color depicts bulk (edge) states. For $B_\parallel < B_{c1} = |t|$, $MZ_2$ is the topological invariant, while for the range $3|t| = B_{c2} \geq B_\parallel \geq B_{c1}$, $CZ_2$ is the topological invariant. In panel (b) we show the creation (black dots) and annihilation (cross) of the Weyl points in reciprocal space for the KM model. In panels (c)-(e), we show the energy spectrum as a function of $k_y$ for three different magnetic fields, corresponding to the three different phases from panel (a): Topological insulator, topological semimetal and trivial insulator.

# 3 Unconventional topological phase transition: Crystalline Weyl semimetal phase

In this section, we explain the phase diagram exhibited by the KM and the BHZ models with PHS and a commuting reflection symmetry when an external magnetic field is applied. Here, the system evolves from a TI, characterized by the topological invariant $MZ_2$, to a trivial insulator. Between these two phases, a Weyl semimetal phase arises for a finite range of magnetic field $B_{c2} \geq B_\parallel \geq B_{c1}$. This phase is protected by the combination of PHS and one of the reflection symmetries: $\tilde{\mathcal{C}} = \mathcal{R}(y)\mathcal{C}$. In the following paragraphs, we will explicitly calculate the topological invariants for the KM model. Then, at the end of this section, we will extend these results to the BHZ model.

## 3.1 Topological insulator phase

In the absence of any other symmetry, the QSH Hamiltonian $H_0^{KM/BHZ}$ always shows a gap opening in the presence of an in-plane Zeeman field $H_Z$ [65]. However, this situation can change in the presence of extra reflection symmetries, which combine with non-local symmetries and can modify the topological classification of the Hamiltonian [70, 71]. In order to account for reflection symmetries in the topological classification, we compute the commutation relations of the reflection operators, $\mathcal{R}(x)$ and $\mathcal{R}(y)$, with the remaining bulk symmetry, i.e. $\mathcal{C}$. Following the notation used in Refs. [70, 71], we assign the labels $R_\pm$ if $\mathcal{R}(i)$ commutes (+) or anticommutes (-) with the particle-hole operator $\mathcal{C}$, i.e. $\mathcal{R}(x) \to R_-$ and $\mathcal{R}(y) \to R_+$.

Table 1: Summary of the topological classifications of both particle-hole symmetric KM and BHZ Hamiltonians taking into account commuting ($R_+$) and anticommuting ($R_-$) reflection symmetries. The first column refers to the case where the system is gapped, while the rest of the columns refers to a gapless Hamiltonian. FS1, FS2 and FS3 reffer to the position of the accidental crossings, or Fermi surfaces in the reciprocal lattice: For FS1, the accidental crossing is placed on the mirror plane and at a high-symmetry point. For FS2, the accidental crossing is on the mirror plane, but away from a high-symmetry point, and for FS3, the accidental crossing is away from both, high-symmetry points and the mirror plane.

| D | Top. insul. | FS1 | FS2 | FS3 |
|---|---|---|---|---|
| $R_+$ | $MZ_2$ | MZ | $MZ_2$ | $CZ_2$ |
| $R_-$ | 0 | 0 | MZ | 0 |

We summarize the topological classification given in Refs. [70,71] in Tab. 1. We can see that for topological insulators (first column) we find that $R_+$ [$\mathcal{R}(y)$], is characterized by a mirror topological invariant $MZ_2$. In turn, $R_-$ [$\mathcal{R}(x)$] exhibits always a trivial phase (0). The rest of the columns will be used below when studying the semimetal phases.

In order to calculate $MZ_2$, we project the 2D Hamiltonian on the 1D reflection-invariant momenta and calculate the topological invariant of the resulting 1D Hamiltonians. At the reflection-invariant momenta [$k_y = 0$ and $k_y = 2\pi/(\sqrt{3}a)$], the effective 1D Hamiltonians commute with $\mathcal{R}(y)$, i.e. $[H_{k_y}(k_x), \mathcal{R}(y)] = 0$. Therefore, it is possible to use the same basis that diagonalizes $\mathcal{R}(y)$, i.e. $U_R \mathcal{R}(y) U_R^\dagger = \mathrm{diag}(\mathbf{1}_{2\times2}, -\mathbf{1}_{2\times2})$, to rewrite $H(k_x, k_y)$ in a block diagonal basis, i.e. $U_R H_{k_y=\mathrm{RIM}}(k_x) U_R^\dagger = \mathrm{diag}\left[ H_{k_y=\mathrm{RIM}}^+(k_x), H_{k_y=\mathrm{RIM}}^-(k_x) \right]$. The superscripts $\pm$ label the reflection parity blocks corresponding to the $\pm 1$ eigenvalues.

The topological invariant of the resulting 1D **D**-class Hamiltonian can be calculated as in the Kitaev model [72]. Thus, we express $H_{k_y=\mathrm{RIM}}^\pm(k_x)$ in the so-called Majorana basis, in which the unitary part of the particle-hole operator $\mathcal{C} = U_c \mathcal{K}$ transforms into $U_c = \mathbf{1}$. At the time-reversal invariant momenta, the Hamiltonian becomes purely imaginary $H_M = iA(k_x)$, where $A(k_x)$ is a real and antisymmetric matrix, $A^T = -A$ [73]. Note that there is only one time-reversal invariant momentum ($k_x = 0$) that together with $k_y = \mathrm{RIM}$ lies inside the first Brillouin zone. Therefore, the invariant is expressed in terms of

$$(-1)^{n_{\mathrm{MZ}_2}^\pm} = \mathrm{Sign}\{\mathrm{Pf}[A_{k_y=0}^\pm(0)]\mathrm{Pf}[A_{k_y=\frac{2\pi}{\sqrt{3}a}}^\pm(0)]\}, \tag{17}$$

with

$$\mathrm{Pf}[A_{k_y=0}^\pm(0)] = \pm B_\parallel + 3t, \tag{18}$$

$$\mathrm{Pf}[A_{k_y=\frac{2\pi}{\sqrt{3}a}}^\pm(0)] = \pm B_\parallel - t. \tag{19}$$

Plugging Eqs. (18) and (19) into Eq. (17), we obtain that $n_{\mathrm{MZ}_2}^\pm$ exhibits a topological insulator phase for $B_\parallel < |t|$. This topological invariant $n_{\mathrm{MZ}_2}^\pm$ characterizes the topological insulating phase until $B_\parallel = B_{c1} = |t|$. For higher magnetic fields, $MZ_2^\pm$ exhibit different values at different reflection parities, indicating a change in the topological invariant of the system [74].

## 3.2 Crystalline Weyl semimetal phase

Setting the magnetic field at $B_{c1} = |t|$ a total of six Weyl points emerge: four placed at the $\Gamma_2, \Gamma_3$-points and two at $\mathbf{K}' = (2\pi/a, 0)$ and $\mathbf{K} = (-2\pi/a, 0)$, see black dots in Fig. 2(b). Increasing further the magnetic field the Weyl points shift distinguishing two different regimes:

$B_{c1} \leq B_{\parallel} \leq \sqrt{5}|t|$ and $\sqrt{5}|t| \leq B_{\parallel} \leq B_{c2}$. For $B_{c1} \leq B_{\parallel} \leq \sqrt{5}|t|$ the crossing points placed at $\mathbf{K}$ and $\mathbf{K}'$ split in the vertical axis into two points. Thus, the Weyl points placed at $\mathbf{K}$ $(\mathbf{K}')$ shift towards the points $(-2\pi/a, \pm 2\pi/\sqrt{3}a)$ $[(2\pi/a, \pm 2\pi/\sqrt{3}a)]$, where they annihilate each other for $B_{\parallel} = \sqrt{5}|t|$.[2] Meanwhile, the crossing points placed at the $\Gamma_2, \Gamma_3$-points evolve slowly towards $\Gamma_0 = (0,0)$, see left panel in Fig. 2(b). In the second regime $(\sqrt{5}|t| \leq B_{\parallel} \leq B_{c2})$, two extra crossing points are created at $\Gamma_1 = (0, \pm 2\pi/\sqrt{3}a)$ and shift vertically towards $\Gamma_0 = (0,0)$, where they annihilate each other together with the remaining 4 crossing points for $B_{\parallel} = 3|t|$, see right panel in Fig. 2(b).

These stable accidental crossing points may appear in a 2D system because of both, the lack of TRS and the presence of PHS and a commuting reflection symmetry. This results effectively in a reduction of the number of allowed Pauli matrices to formulate the Hamiltonian. We can understand this by writing the eigenenergies of Eq. (2) [and Eq. (10)] in the presence of a Zeeman term, that is

$$E_{\pm} = \pm\sqrt{\left(\sqrt{f_x^2 + f_y^2} - B_{\parallel}\right)^2 + f_z^2}, \tag{20}$$

which correspond to the eigenenergies from a Hamiltonian containing only two Pauli matrices, and therefore, there are only two conditions for the closing of the gap: $f_x^2 + f_y^2 = B_{\parallel}^2$ and $f_z = 0$. Since the number of independent parameters exceeds the number of conditions, we expect to close the gap for a finite range of magnetic fields.

These accidental crossings lie in general off high-symmetry points and outside the reflection planes (FS3, see Tab. 1) and thus, alone, the reflection symmetry cannot protect Fermi surfaces. However, a combination of PHS and reflection symmetry, i.e. $\tilde{\mathcal{C}} = \mathcal{R}(y)\mathcal{C}$ does [74]. This operator transforms the Hamiltonian as

$$\tilde{\mathcal{C}}H(k_x, k_y)\tilde{\mathcal{C}}^{-1} = -H(-k_x, k_y). \tag{21}$$

Therefore, $\tilde{\mathcal{C}}$ can be understood as an effective PHS acting within the 1D Hamiltonian, where $k_y$ is treated as an extra parameter of the model. For this reason, we can define a topological invariant $n_{CZ_2}$ analogous to the Kitaev model that is $k_y$-dependent. This topological invariant can only change across the gap-closing points, and gives rise to Fermi arc states that connect two Fermi points. Thus, we first write the Hamiltonian in the Majorana basis, i.e. the basis that transforms $\tilde{\mathcal{C}}' = \mathcal{K}$. In this basis, the Hamiltonian becomes antisymmetric at the $x-$reflection-invariant momenta $k_x = 0$ and $2\pi/a$, i.e. $H'(k_x = RIM) = iA_{k_x=RIM}(k_y)$, which allows to define the topological invariant $n_{CZ_2}$ as

$$(-1)^{n_{CZ_2}(k_y)} = \text{Sign}\{\text{Pf}[A_{k_x=0}]\text{Pf}[A_{k_x=\frac{2\pi}{a}}]\}, \tag{22}$$

with

$$\text{Pf}[A_{k_x=0}] = B_{\parallel}^2 - 5t^2 - 4t^2 \cos(\sqrt{3}k_y/2a), \tag{23}$$

$$\text{Pf}[A_{k_x=\frac{2\pi}{a}}] = B_{\parallel}^2 - 5t^2 + 4t^2 \cos(\sqrt{3}k_y/2a). \tag{24}$$

This topological invariant sets the conditions to observe Fermi arcs connecting the Weyl points described above. In Fig. 3 we evaluate numerically Eq. (22) and show the extension of the Fermi arc as a function of $B_{\parallel}/|t|$ and $k_y a$. From these results, we observe that the Fermi arcs appear centered at $k_y = 0$ for $B_{\parallel} = B_{c1}$, connecting the Weyl points that appear around $\mathbf{K}' = (2\pi/a, \pm\delta)$ and $\mathbf{K} = (-2\pi/a, \pm\delta)$. Here, $\delta$ goes from 0 up to $2\pi/\sqrt{3}a$ as $B_{\parallel}$ goes from $|t|$ up to $\sqrt{5}|t|$, where the Fermi arcs extend over the whole reciprocal lattice. For $B_{\parallel} = \sqrt{5}|t|$, the Fermi arcs connect the new Weyl points created at $\Gamma_1$. Increasing $B_{\parallel}$ further, $n_{CZ_2}$ predicts the annihilation of these Weyl points at $k_y = 0$ for $B_{\parallel} = 3|t|$.

---

[2]Due to the periodicity of the reciprocal space, $(\pm 2\pi/a, 2\pi/\sqrt{3}a)$ and $(\pm 2\pi/a, -2\pi/\sqrt{3}a)$ are equivalent.

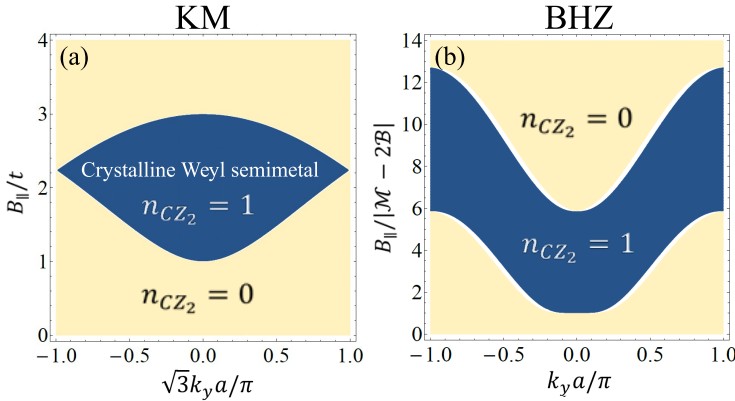

Figure 3: Panels (a) and (b) show the phase diagrams of $n_{CZ_2}$ as a function of $k_y$ and $B_{\parallel}$ for the KM and BHZ model, respectively. Blue areas correspond to $n_{CZ_2} = 1$, while the yellow areas to $n_{CZ_2} = 0$. Thus, blue areas show the extension of the Fermi arcs in reciprocal space for a given $B_{\parallel}$. In the BHZ model we used $\mathcal{A} = 182.25$ meV, $\mathcal{B} = 343$ meV and $\mathcal{M} = 586$ meV.

We confirm these results numerically in Fig. 2(a), where we show the energy spectrum of an AC nanoribbon as a function of the in-plane magnetic field $B_{\parallel}$ at $k_y = 0$. Consistent with the calculation of the topological invariants, we observe the presence of zero energy states for $B_{\parallel} \leq 3|t|$, see Fig. 3. Then, in Figs. 2(c) and (d), we show three representative examples of the energy dispersion in each phase: In panel (c), $B_{\parallel} = 0.4|t|$ and the energy dispersion is that of a topological insulator. Panel (d) with $B_{\parallel} = 1.4|t|$ shows a crystalline Weyl semimetal and panel (e) with $B_{\parallel} = 3.4|t|$ a trivial insulator.

It is important to remark at this point that until now we have discussed the symmetries and topological invariants of the bulk Hamiltonian, and therefore, these arguments apply in principle to both ZZ and AC boundary conditions. However, one has to realize that ZZ boundary condition does not preserve the reflection symmetry $\mathcal{R}(y)$. Thus, we expect that ZZ nanoribbons always exhibit a trivial phase in the presence of magnetic fields. Conversely, AC boundary conditions preserve both reflection symmetries $\mathcal{R}(x)$ and $\mathcal{R}(y)$. In the next section, we will confirm these symmetry arguments by diagonalizing ZZ and AC nanoribbons in the presence of an in-plane magnetic field and couplings that break PHS.

## 3.3 Extension of the analysis to the BHZ model

A similar topological phase transition is observed in the particle-hole symmetric BHZ model with a good quantum number $k_y$. In this case, its origin is quite surprising and counterintuitive because the geometry of the BHZ model is that of a square lattice and thus, one would expect a similar behavior along both boundaries. However, there is a mathematical difference between both boundaries, which arises due to the TRS of the BHZ Hamiltonian, see Eq. (10). Under this condition, different pseudospin Pauli matrices $\sigma_x$ (real) and $\sigma_y$ (imaginary) acquire a different spin functional form $s_z$ and $s_0$, respectively. Then, because of this difference, the reflection operators acquire a different pseudospin functional form, i.e. $\mathcal{R}(x) = \sigma_0 s_\theta$, while $\mathcal{R}(y) = \sigma_z s_\theta$, which gives rise to different commutation relations between $\mathcal{R}(x/y)$ and $\mathcal{C}$, i.e. $\mathcal{R}(x) \rightarrow R_-$ and $\mathcal{R}(y) \rightarrow R_+$. Now, only the infinite mass boundary condition $M_x = \sigma_z$ (with good quantum number along the $y$-direction) commutes with $\mathcal{R}(x)$ and $\mathcal{R}(y)$, and therefore, all the topological properties obtained for AC edge states in the KM model apply to

the BHZ model for finite strips that are infinite along the $y$ direction.

Here, the TI phase extends up to Zeeman energies of $B_\parallel < |\mathcal{M} - 2\mathcal{B}|$. As the Zeeman energies are further increased, the system enters the crystalline WSM phase. This is also illustrated by Fig. 3(b), which shows the corresponding topological invariant $n_{\text{CZ}_2}$ computed from Eq. (22) and

$$\text{Pf}[A_{k_x=0}] = B_\parallel^2 - \big(\mathcal{B} - \mathcal{M} + \mathcal{B}\cos(k_y a)\big)^2 - \mathcal{A}^2\sin^2(k_y a),$$

$$\text{Pf}[A_{k_x=\frac{2\pi}{a}}] = B_\parallel^2 - \big(\mathcal{B} + \mathcal{M} - \mathcal{B}\cos(k_y a)\big)^2 - \mathcal{A}^2\sin^2(k_y a).$$

In this case, a pair of Weyl points emerges at the $\Gamma$ point in the bulk spectrum. With increasing $B_\parallel$, these Weyl points split and move along the $k_y$ axis: One Weyl point tends to $k_y = \pi/a$, while the other tends to $k_y = -\pi/a$. At $k_y = \pm\pi/a$, the two Weyl point merge and are annihilated.

Moreover, additional Weyl points appear as $B_\parallel$ is increased, similar to the KM model. For the BHZ model, however, the number and position of the additional Weyl points in the bulk spectrum depends on the ratio between $\mathcal{A}$ and $\mathcal{B}$. The ratio $\mathcal{B}/\mathcal{A}$ also determines the extent of the crystalline WSM phase: In the typical case of $|\mathcal{A}| < |\mathcal{B}|$, the semimetal phase exists for $|\mathcal{M} - 2\mathcal{B}| < B_\parallel < |\mathcal{M} + 2\mathcal{B}|$, which is also the case shown in Fig. 3(b). If $|\mathcal{A}| > |\mathcal{B}|$ (not shown here), the semimetal phase exists for $|\mathcal{M} - 2\mathcal{B}| < B_\parallel < \sqrt{\mathcal{A}^2(\mathcal{A}^2 + \mathcal{M}^2 + 2\mathcal{M}\mathcal{B})/(\mathcal{A}^2 - \mathcal{B}^2)}$. In this case, the transition from the semimetal phase to the trivial phase occurs at a Zeeman energy larger compared to the case of $|\mathcal{A}| < |\mathcal{B}|$.

# 4 Breaking particle-hole symmetry: Estimation of the effective g-factor

In the presence of PHS breaking terms, the topological states are no longer protected and an external magnetic field is, in principle, able to open a gap. We characterize the stability of the removed protection through the effective $g^*$ factor, which is the ratio between the gap

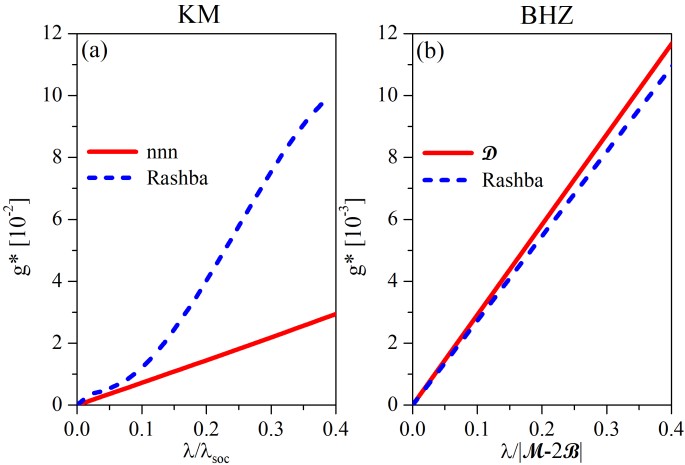

Figure 4: (Color online) Effective $g$-factors as a function of the PHS breaking term parameter $\lambda$ for the KM [panel (a)], we used next nearest neighbor and Rashba, while for the BHZ model [panel (b)], the assymetric mass $\mathcal{D}$ and the Rashba spin orbit coupling. For the parameters used in the calculation, we refer to the main text. In the BHZ model we used $\mathcal{A} = 182.25\,\text{meV}$, $\mathcal{B} = 343\,\text{meV}$ and $\mathcal{M} = 676\,\text{meV}$.

opened ($\Delta_i$) in the (previously protected) AC or $y$ directions, and gaps $\Delta_\perp$ opened in the (unprotected) ZZ or $x$ directions, i.e. $g^* = \Delta_i / \Delta_\perp$.

In Fig. 4 we show $g^*$ as a function of the PHS breaking parameter, generically called $\lambda$. In general, this value changes slightly with the applied magnetic field, thus, we average $g^*$ over a range of magnetic fields. We have used the PHS breaking terms introduced in Sec. 2, i.e. in the KM model we have used the Rashba and next-nearest neighbor Hamiltonians. Then, for the BHZ model, the asymmetric Zeeman, Rashba and asymmetric masses. As expected, the numerical results for $g^*$ confirm the topological invariant analysis from the previous section: $g^* = 0$ for $\lambda = 0$ with $g^*$ increasing proportionally to $\lambda$. Furthermore, a direct comparison between $g^*$ in the KM and BHZ models reveals no further difference, i.e. $g^*_{\text{BHZ}} \sim g^*_{\text{KM}}$. However for the experimentally relevant parameters, the BHZ model exhibits a larger particle-hole asymmetry ($\lambda/|\mathcal{M} - 2\mathcal{B}| \sim 25$) than in the KM model ($\lambda/\lambda_{\text{SOC}} \sim 0.1$). Therefore, we expect to observe almost perfectly protected edge states, reminiscent of this topological crystalline protection for the KM model, while for the BHZ model there are almost no difference between both boundaries, i.e. $g^*_{\text{BHZ}} \approx 1$ for realistic parameters.

In the KM model, the PHS breaking term that gives rise to a larger gap is the Rashba SOC, given in Eq. (8). As we mentioned above, in the worst scenario the magnetic field can mix counterpropagating modes giving rise to the opening of a gap, see Fig. 5 (a) and (b). However, there is an anisotropic effect of the Zeeman splitting coming from the fact that to lowest order in momentum $k_y$, the reflection symmetry $\mathcal{R}(y)$ is conserved [42]. Thus, when the magnetic field is parallel to the AC boundary ($B_y$ or $\theta = \pi/2$) the gap does not open and it is in principle possible to observe the topological phase transition, see Fig. 5 (c) and (d).

# 5 Conductance in the presence of disorder

In addition to the previously studied PHS breaking terms, we now explore the impact of disorder and a Zeeman field on the quantized conductance $G = 2e^2/h$ of the helical edge states. The presence of disorder in crystalline topological insulators has a negligible impact on their

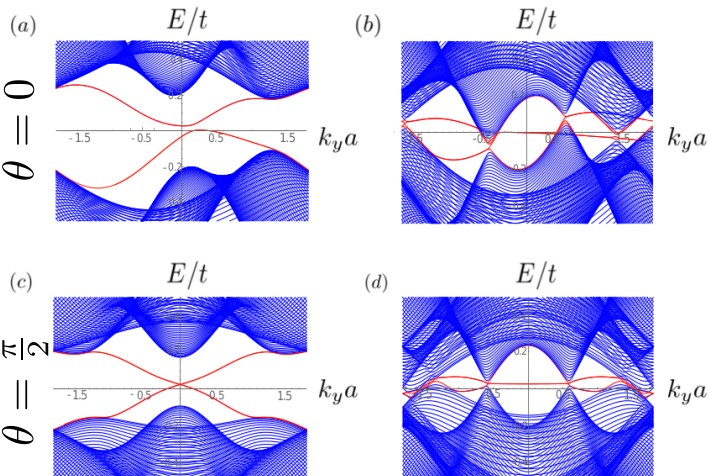

Figure 5: Energy spectra of AC nanoribbons in the presence of Rashba SOC and an applied magnetic field in $x$-direction ($\theta = 0$) [(a) and (b) panels] and $y$-direction ($\theta = \pi/2$) [(c) and (d) panels]. We used $B_\parallel = 0.8|t|$ in the topological insulator phase, and $B_\parallel = 1.2|t|$ in the crystalline Weyl semimetal phase.

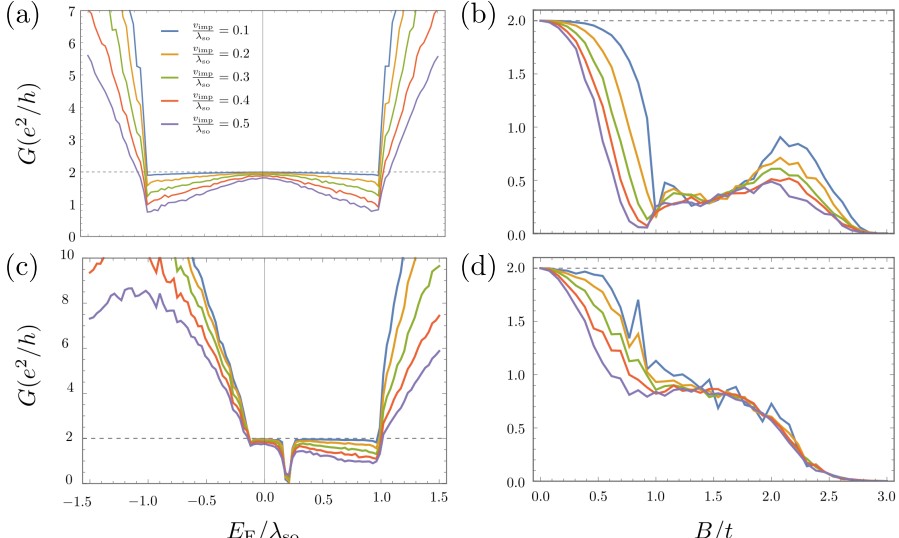

Figure 6: Linear conductance for the Kane-Mele model as a function of the Fermi energy with $B = 0.25t$ (left panels) and as a function of the Zeeman energy $B/t$ with $E_\mathrm{F} = 0.0$ (right panels) with different disorder strengths $v_\mathrm{imp}/\lambda_\mathrm{so} = 0.1, 0.2, 0.3, 0.4, 0.5$. In the top panels, the calculations are performed for $\lambda_\mathrm{R} = 0.0$, while for the bottom panels $\lambda_\mathrm{R} = 0.15t$. All conductance calculations are the result of the average over 100 realizations in a disordered slab of length $150a_0$.

topological protection because, in average, reflection symmetries are conserved [71]. However, in our case, the presence of disorder also breaks particle-hole symmetry and therefore, it is necessary to check its impact on the quantized conductance. To this aim, we use standard Green's function methods, and calculate the linear conductance of a system consisting of two semi-infinite leads [75] coupled through a finite scattering sector where we introduce disorder through a local random potential $v_\mathrm{imp}(x, y)$, with maximum value of $|v_\mathrm{imp}|$. Here, the linear conductance at zero temperature is given by the well-known expression

$$G = \frac{4e^2}{h} \mathrm{Tr}[\Gamma_L G_\mathrm{c}^r \Gamma_R G_\mathrm{c}^a],\tag{25}$$

where $G_\mathrm{c}^{r,a}$ are the retarded and advanced Green's functions for the section $c$. Note that due to current conservation, the calculations do not depend on the chosen section $c$. Moreover, $\Gamma_{L,R} = 2\mathrm{Im}\{\Sigma_{L,R}^a\}$ are the imaginary parts of the left- and right-self-energies $\Sigma_{L,R}^a = V_\mathrm{CL} g_\mathrm{LL}^{r,a} V_\mathrm{LC}$, and the terms $V_\mathrm{CL/R,L/RC}$ represent the tunnel couplings between the left/right (right/left) and the section $c$. Besides, the trace runs over the transversal direction of the junction.

In Fig. 6 we show the linear conductance for the KM model model as a function of the Fermi energy (a,c) and the Zeeman energy (b,d) for $\lambda_\mathrm{R} = 0$ (a,b) and $\lambda_\mathrm{R} = 0.15t$ (c,d). In the absence of Rashba spin-orbit coupling (a,b), we observe that for small disorder strength relative to the insulating gap $\lambda_\mathrm{so}$, i.e. $v_\mathrm{imp}/\lambda_\mathrm{so} \sim 0.1$, the conductance remains approximately quantized $G = 2e^2/h$ for $B < B_\mathrm{c1}$. Note that this is not necessarily a small disorder strength, because materials like bismuthene exhibit a gap of the order $\lambda_\mathrm{so} \sim 1\,\mathrm{eV}$ [44, 45]. For larger disorder strengths, the conductance reduces without the opening of a gap. Furthermore, we note that the conductance becomes more reduced the closer the Fermi energy is to the bulk states. Thus, presumably the lack of protection comes from the coupling between the edge and bulk states. This is in contrast to the impact caused by previously studied terms that break

particle-hole symmetry, which couple counterpropagating states giving rise to a gap at the crossing point, with the consequent drop in conductance, see Fig. 6(c).

In Figs. 6 (b) and (d) we show the linear conductance for $E_F = 0$ as a function of the Zeeman energy in the absence and presence of Rashba spin-orbit coupling, (b) and (d) respectively. For magnetic fields above $B > B_{c1} = |t|$, the conductance at $E_F = 0$ drops since the semimetallic phase does not exhibit a quantized longitudinal conductance. In order to explore the stability of the semimetallic phase one needs to perform Hall conductance calculations, which are beyond the scope of this contribution.

# 6    Potential realization in topoelectrical circuits

Above, we have shown that the combination of PHS and reflection symmetry along a given edge gives rise to an unconventional phase transition in both, the KM or BHZ models. Here, a Weyl semimetal phase separates a crystalline topological insulator phase from a trivial phase. There are, however, several issues which prevent this transition from being observed experimentally: First, this transition occurs in systems with PHS, which in real systems is broken, for example, by disorder (see Sec. 5 above), by the asymmetry between conduction and valence band effective masses in the BHZ model, or next-nearest neighbor hopping in the KM model. Second, even for particle-hole symmetric systems the Zeeman energies required to enter the Weyl semimetal phase are huge, corresponding to magnetic fields of $10^3$-$10^4$ T for typical g factors.

While it is thus unlikely to observe such transitions in real systems, one could test our predictions in novel metamaterials such as topoelectrical circuits. The recently demonstrated topoelectrical circuits, for example, allow one to transfer and study topological concepts from quantum-mechanical to classical electronic systems [76–79]. Here, circuits consisting of resistor, inductor and capacitor components are described by a circuit Laplacian, similar to the Hamiltonian of a quantum mechanical system [76]. By engineering such circuits appropriately, one can thus realize the particle-hole symmetric cases of the KM and BHZ models studied here and also mitigate the detrimental effects of disorder. Likewise, the circuits allow for an implementation of the topoelectrical equivalents to huge Zeeman energies. Another class of metamaterials that offers the possibility of observing the phenomena predicted in this manuscript are optical lattices of cold atoms, which can be used to mimick condensed matter systems [80–82].

# 7    Conclusions

We have studied an unconventional topological phase transition in QSH systems in an in-plane magnetic field. This phase transition consists of three different phases: a topological insulator phase, a crystalline Weyl semimetal phase, and a trivial phase. The key point to induce this topological phase transition is the presence of a topological crystalline protection, which prevents counter-propagating modes from mixing when a magnetic field is applied. This protection appears through the combination of PHS and a commuting reflection symmetry. We have calculated the topological invariants within Kane-Mele and BHZ models for QSH edge states and crystalline Weyl semimetal. Moreover, we have studied the stability of the crystalline protection, when adding particle-hole symmetry breaking terms. We find that the crystalline edge state protection and the unconventional topological phase transition is robust for the KM model. The discussed protecion might be manifested in transport measurements [42,83] and can also be important for generating Majorana modes [84]. Since the predicted magnetic fields to observe crystalline Weyl semimetal phase are quite high, we believe that one of new venues where this phase might be easier to detect are topological circuits [76].

## Acknowledgments

We thank M. Kharitonov, Song-Bo Zhang, W. Beugeling, C. de Beule and R. Thomale for valuable discussions. This work was supported by the German Science Foundation (DFG) via Grant No. SFB 1170 "ToCoTronics", through the Würzburg-Dresden Cluster of Excellence on Complexity and Topology in Quantum Matter-ct.qmat (EXC 2147, project-id 390858490) and by the ENB Graduate School on Topological Insulators.

## A Discretization

In order to check our arguments about the different phases and their topological invariants determined from the bulk Hamiltonians given in Secs. 2.1 and 2.2, we also compute the energy spectra of nanoribbons and finite strips, respectively. The Hamiltonians for these can be obtained by preforming 1D Fourier transformations of the bulk Hamiltonians from Secs. 2.1 and 2.2.

For the KM model and our choice of basis vectors, $\mathbf{a}_1 = a\mathbf{e}_x$ and $\mathbf{a}_2 = -(a/2)\mathbf{e}_x + (\sqrt{3}a/2)\mathbf{e}_y$, a Fourier transform with respect to $\sqrt{3}k_y/2$ yields nanoribbons with ZZ edges along the $x$ direction, a finite number of lattice sites along the $y$ direction, and a good momentum quantum number $k_x$. By a Fourier transform with respect to $k_x/2$, on the other hand, we obtain nanoribbons with AC edges along the $y$ direction, a finite number of lattice sites along the $x$ direction, and a good momentum quantum number $k_y$. The ZZ and AC directions and their corresponding coordinate axes are shown in Fig. 1(a).

Similarly, for the BHZ model a finite strip with a finite width in $y$ direction and a good quantum number $k_x$ can be obtained by a Fourier transform of the $\mathbf{k}$-dependent bulk Hamiltonian with respect to $k_y$. Conversely, Fourier transforming the bulk Hamiltonian with respect to $k_x$ yields a finite strip with a good quantum number $k_y$ and a finite width in $x$ direction. The directions are depicted in Fig. 1(c). In all calculations we have used a lattice constant of $a = 2\,\text{nm}$ and $N = 299$ sites, resulting in a ribbon of $600\,\text{nm}$.

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
