# Peer review of "Crystalline Weyl semimetal phase in Quantum Spin Hall systems under magnetic fields"

_SciPost Physics Core, doi:SciPost Phys. Core 5, 024 (2022)_

## Round 1 · Referee Report · Anonymous · 2019-6-17

Strengths

1-The authors give a concrete tight-binding model that exhibits a Weyl semimetal phase in two dimensions, protected by crystalline and particle-hole symmetries.
2-By contrast with previous work, this phase is stable against the breaking of time-reversal (T) symmetry, and is in fact induced by a T-breaking perturbation, an applied in-plane magnetic field.
3-The phase is moderately stable against some perturbations that break particle-hole symmetry.
4-The model considered can potentially describe real materials, such as HgTe/CdTe quantum wells and bismuthene on SiC.

Weaknesses

1-The restriction to particle-hole symmetric Hamiltonians for a band insulator/semiconductor model (as opposed to a superconducting one) seems artificial.
2-Although small deviations from particle-hole symmetry are studied (Sec. 4 of the paper), it is not clear why the authors only consider the Rashba and next-nearest-neighbor hopping terms on the honeycomb lattice. Is there a specific reason why only these two terms are considered?
3-The magnetic fields required to observe the crystalline Weyl phase are such that the associated Zeeman energy would be on the order of the bandwidth, which would correspond to fields on the order of 10$^3$ - 10$^4$ T. Thus it appears very unlikely that this phase could be observed in practice.

Report

The idea of applying an in-plane magnetic field to induce a 2D Weyl phase is interesting, and the authors do a good job of characterizing the Hamiltonian they are proposing, by studying the bulk/edge spectrum as well as computing topological invariants protected by various symmetries.

My main concern is how relevant this study is to real materials. Is there an advantage to realizing an unconventional TPT using an external magnetic field, as opposed to the inversion-breaking proposal of Ref. 37 where strain/electric fields are used? Could the authors elaborate on the idea in their last sentence, i.e., how the high magnetic field requirements could be circumvented by using topoelectrical circuits?

Also, I think the authors should better justify their choice of particle-hole symmetry breaking terms: are these terms energetically dominant for specific materials (or classes of materials)? For HgTe/CdTe, their Hamiltonian Eq. (10) does not contain bulk inversion asymmetry terms which are important for some of the physics of the quantum spin Hall state in this system. Do these affect their analysis from a symmetry point of view? Also, the simplest example of particle-hole symmetry breaking term, not discussed in the manuscript, is an on-site potential, as induced for example by random impurities. Can the authors comment on the effect of disorder on their conclusions regarding the stability of the crystalline 2D Weyl phase in their model?

A few minor issues:
- The authors use both "lattice constant $a$" and "step size $a$"; the latter is a bit confusing, and it would probably be best to use "lattice constant" throughout.
- Similarly, they sometimes use $i$ and sometimes $\iota$ for the imaginary unit.
- In the penultimate paragraph on p. 9, the first sentence contains a reference to Fig. 2(b), but I believe it should be Fig. 2(a) (?)
- p. 4, paragraph after Eq. (3): "magnetic field module"; the authors probably mean modulus (or magnitude, or amplitude)?
- Top of p. 12: "anysotropic" $\rightarrow$ "anisotropic"

---

## Round 1 · Referee Report · Anonymous · 2019-7-2

Strengths

1-The paper is well-structured and clearly written
2-The authors study two different models of a QSH insulator, thus making their results potentially applicable to a wider variety of systems
3-The authors study the effect of some PHS-breaking terms, which are expected to be present in real materials and break the protection of the topological phase

Weaknesses

1-I believe that both the topological insulator and the topological semimetal phase are unstable against generic disorder. The latter should lead to a localization of edge and bulk modes, hindering the potential observation of this phase in a traditional, condensed-matter setup.

Report

The authors study the effect of an in-plane Zeeman field on a QSH insulator, considering both the Kane-Mele and the Bernevig-Hughes-Zhang models. Although time-reversal symmetry is broken, the system remains topological due to the presence of particle-hole symmetry as well as mirror symmetry. Further upon increasing the magnetic field, the system evolves from a topological insulator to a trivial phase via an intermediate, topological semimetal phase, a scenario the authors refer to as an "unconventional topological phase transition."

The paper is well-structured and easy to follow. The authors characterize both the KM and BHZ models in a pedagogical, step-by-step manner, identifying the symmetries required for topological protection. Further, they also study the effect of PHS breaking terms, showing that for certain parameter regimes relevant to condensed-matter realizations of the QSHE, the topological insulator/semimetal phase may remain robust.

My main criticism of this work comes from the fact that I believe the topological insulator and semimetal phase to be unstable against disorder, hindering their potential observation in a conventional condensed-matter experiment. The models break TRS, so one would expect that generic disorder will lead to weak localization in two dimensions. I suspect that, even if the Rashba and NNN terms are not added, simply adding random disorder in the chemical potential and hopping terms will localize both the edge modes of the topological insulator and the bulk modes of the topological semimetal, rendering both phases trivial. This is in contrast to Ref. 37, where the topological semimetal is expected to remain conducting when weak disorder is present. This is due to the fact that TRS is intact in the model of Ref. 37, so weak anti-localization is possible in 2D.

I suggest that the authors add a section studying disorder in their system, showing, for instance, its effect on the conductance of edge and bulk modes. Further, the authors should comment more on potential realizations of their model in setups where the effects of disorder can be mitigated. These may include ultracold atoms or topoelectric circuits, which are mentioned only very briefly at the end of the paper.

Requested changes

1-Add a section studying the effect of disorder on the edge and bulk states of the topological insulator and the topological semimetal
2-Expand the discussion on meta-material realizations as potential ways of mitigating the effect of disorder
2-The authors introduce many abbreviations throughout the text, such as HSP, TRIM, RIM to refer to points in momentum space. The paper would be clearer if some of these are removed and the terms spelled out in full.
3-On page 7, I do not believe that the invariant characterizing the topological insulator should be called a mirror "Chern number." Chern numbers are integer and not Z2, and typically computed on even-dimensional (sub-)manifolds, whereas the authors consider 1D slices of the Brillouin zone. This terminology should be clarified in order to avoid confusion.

---

## Round 2 · Author Response

Reply to Referee 1:

Strengths 1-The authors give a concrete tight-binding model that exhibits a Weyl semimetal phase in two dimensions, protected by crystalline and particle-hole symmetries. 2-By contrast with previous work, this phase is stable against the breaking of time-reversal (T) symmetry, and is in fact induced by a T-breaking perturbation, an applied in-plane magnetic field. 3-The phase is moderately stable against some perturbations that break particle-hole symmetry. 4-The model considered can potentially describe real materials, such as HgTe/CdTe quantum wells and bismuthene on SiC.

...

The idea of applying an in-plane magnetic field to induce a 2D Weyl phase is interesting, and the authors do a good job of characterizing the Hamiltonian they are proposing, by studying the bulk/edge spectrum as well as computing topological invariants protected by various symmetries.

We thank the Referee for her/his time in reviewing our manuscript and are gratified about her/his overall positive assessment of our work.

Weaknesses 1-The restriction to particle-hole symmetric Hamiltonians for a band insulator/semiconductor model (as opposed to a superconducting one) seems artificial. ... 3-The magnetic fields required to observe the crystalline Weyl phase are such that the associated Zeeman energy would be on the order of the bandwidth, which would correspond to fields on the order of 103-104 T. Thus it appears very unlikely that this phase could be observed in practice.

...

My main concern is how relevant this study is to real materials. Is there an advantage to realizing an unconventional TPT using an external magnetic field, as opposed to the inversion-breaking proposal of Ref. 37 where strain/electric fields are used? Could the authors elaborate on the idea in their last sentence, i.e., how the high magnetic field requirements could be circumvented by using topoelectrical circuits?

As the Referee correctly points out, it is unlikely that the unconventional topological phase transition described in our manuscript can be realized in real materials because of the extremely large magnetic fields required. Likewise, the presence of particle-hole asymmetry in real systems (if one does not go to systems with superconductivity) also prevents this transition from being easily observed in real materials.

Instead, we envision that one could test our predictions in novel metamaterials such as topoelectrical circuits. The recently demonstrated topoelectrical circuits, for example, allow one to transfer and study topological concepts from quantum-mechanical to classical electronic systems [see Communications Physics 1, 1 (2018), Nat. Phys. 14, 925 (2018), PNAS 118 32 (2021), and Nat. Phys. 16, 747 (2020) (Refs. 75-78 in the revised manuscript)]. In these setups, circuits consisting of resistor, inductor and capacitor components are described by a circuit Laplacian, similar to the Hamiltonian of a quantum mechanical system. By engineering such circuits appropriately, one can thus realize the particle-hole symmetric cases of the KM and BHZ models studied here with large Zeeman energies and also mitigate the detrimental effects of disorder (discussed below). Similarly, optical lattices of cold atoms, which can be used to mimic condensed matter systems [Advances in Physics 56, 243 (2007), Rev. Mod. Phys. 80, 885 (2008), Nature 462, 74 (2009) (Refs. 79-81 in the revised manuscript)], offer the possibility of observing the phenomena predicted in this manuscript.

We have added a new Sec. 6 in the revised manuscript that discusses those topoelectrical circuits in more detail.

2-Although small deviations from particle-hole symmetry are studied (Sec. 4 of the paper), it is not clear why the authors only consider the Rashba and next-nearest-neighbor hopping terms on the honeycomb lattice. Is there a specific reason why only these two terms are considered?

...

Also, I think the authors should better justify their choice of particle-hole symmetry breaking terms: are these terms energetically dominant for specific materials (or classes of materials)? For HgTe/CdTe, their Hamiltonian Eq. (10) does not contain bulk inversion asymmetry terms which are important for some of the physics of the quantum spin Hall state in this system. Do these affect their analysis from a symmetry point of view? Also, the simplest example of particle-hole symmetry breaking term, not discussed in the manuscript, is an on-site potential, as induced for example by random impurities. Can the authors comment on the effect of disorder on their conclusions regarding the stability of the crystalline 2D Weyl phase in their model?

For the honeycomb lattice, we consider Rashba and next-nearest-neighbor hopping terms as examples for particle-hole asymmetric terms because they result in larger gap openings than other terms, such as those that include an energy difference between the AB sublattice, sigma_z, typical for non-planar hexagonal lattices, like silicene.

For HgTe/CdTe, bulk inversion asymmetry (BIA) of strength is described by a Hamiltonian , where and are Pauli matrices in electron/hole space and spin space, respectively. This BIA term preserves particle-hole symmetry and, for finite strips that are infinite in the y direction (called a ‘y strip’ in the following), it also preserves the reflection symmetries. Therefore, the topological crystalline insulator states in y strips remain protected in the presence of BIA as long as . This is illustrated by Fig. R1, where the energy spectrum of y strips remains gapless for finite (particle-hole symmetric) Zeeman terms. In the revised manuscript, we have added a sentence mentioning the role of BIA.

Fig. R1: Energy spectrum at for different directions of the magnetic field and different finite strips with a width of 300 lattice sites: strips, i.e. strips that are infinite in the direction with a good momentum quantum number and finite in the direction, and strips, i.e. strips that are infinite in the direction with a good momentum quantum number and finite in the direction. Here, the Zeeman term is particle-hole symmetric, and , and is the absolute value of the Zeeman field, Moreover, we have chosen . The other parameters are , and (similar to Fig. 4 in the main text).

To address the Referee’s comment on disorder, we have added a new Sec. 5 where we study the two-terminal conductance in the presence of disorder for the Kane-Mele model. These calculations show that for small disorder strengths νimp~0.1 λso, the quantized conductance is only slightly modified for all magnetic fields in the topological insulating regime. In contrast to other terms that break PHS, like Rashba, conductance calculations in the presence of disorder show a lack of a gap opening. Instead, in the presence of disorder and magnetic fields we observe a reduction of the conductance that increases as the Fermi energy becomes closer to the bulk states, which supports the idea that the lack of quantization comes from the coupling to the bulk states. Similar conclusions can be drawn for the BHZ model. Furthermore, we have not analyzed the Hall conductance for the Weyl semimetal phase, although we believe that similar conclusions can be found in that case.

A few minor issues: - The authors use both "lattice constant a " and "step size a "; the latter is a bit confusing, and it would probably be best to use "lattice constant" throughout. - Similarly, they sometimes use i and sometimes ι for the imaginary unit. - In the penultimate paragraph on p. 9, the first sentence contains a reference to Fig. 2(b), but I believe it should be Fig. 2(a) (?) - p. 4, paragraph after Eq. (3): "magnetic field module"; the authors probably mean modulus (or magnitude, or amplitude)? - Top of p. 12: "anysotropic" → "anisotropic"

We would also like to thank the Referee for pointing out several minor issues and typos, all of which have been addressed or corrected in the revised manuscript.

With the changes to the manuscript made to account for the issues raised by both Referees, we believe that our revised manuscript is now suitable for publication in SciPost. Reply to Referee 2:

Strengths 1-The paper is well-structured and clearly written 2-The authors study two different models of a QSH insulator, thus making their results potentially applicable to a wider variety of systems 3-The authors study the effect of some PHS-breaking terms, which are expected to be present in real materials and break the protection of the topological phase

...

The authors study the effect of an in-plane Zeeman field on a QSH insulator, considering both the Kane-Mele and the Bernevig-Hughes-Zhang models. Although time-reversal symmetry is broken, the system remains topological due to the presence of particle-hole symmetry as well as mirror symmetry. Further upon increasing the magnetic field, the system evolves from a topological insulator to a trivial phase via an intermediate, topological semimetal phase, a scenario the authors refer to as an "unconventional topological phase transition."

The paper is well-structured and easy to follow. The authors characterize both the KM and BHZ models in a pedagogical, step-by-step manner, identifying the symmetries required for topological protection. Further, they also study the effect of PHS breaking terms, showing that for certain parameter regimes relevant to condensed-matter realizations of the QSHE, the topological insulator/semimetal phase may remain robust.

We thank the Referee for her/his time in reviewing our manuscript and are gratified about the overall positive assessment of our work.

Weaknesses 1-I believe that both the topological insulator and the topological semimetal phase are unstable against generic disorder. The latter should lead to a localization of edge and bulk modes, hindering the potential observation of this phase in a traditional, condensed-matter setup.

...

My main criticism of this work comes from the fact that I believe the topological insulator and semimetal phase to be unstable against disorder, hindering their potential observation in a conventional condensed-matter experiment. The models break TRS, so one would expect that generic disorder will lead to weak localization in two dimensions. I suspect that, even if the Rashba and NNN terms are not added, simply adding random disorder in the chemical potential and hopping terms will localize both the edge modes of the topological insulator and the bulk modes of the topological semimetal, rendering both phases trivial. This is in contrast to Ref. 37, where the topological semimetal is expected to remain conducting when weak disorder is present. This is due to the fact that TRS is intact in the model of Ref. 37, so weak anti-localization is possible in 2D.

I suggest that the authors add a section studying disorder in their system, showing, for instance, its effect on the conductance of edge and bulk modes...

...

Requested changes 1-Add a section studying the effect of disorder on the edge and bulk states of the topological insulator and the topological semimetal

As mentioned in the reply to the first referee, we have performed additional conductance calculations for the Kane-Mele model in the presence of disorder. These calculations show that for small disorder strengths νimp~0.1 λso, the quantized conductance is only slightly modified for all magnetic fields in the topological insulating regime. In contrast to other terms that break PHS, like Rashba, conductance calculations in the presence of disorder show a lack of a gap opening. Instead, in the presence of disorder and magnetic fields we observe a reduction of the conductance that increases as the Fermi energy becomes closer to the bulk states, which supports the idea that the lack of quantization comes from the coupling of the edge and bulk states. Note that, although we have focused on the Kane and Mele model, similar conclusions can be drawn for the BHZ model. Furthermore, we have not analyzed the Hall conductance for the Weyl semimetal phase, although we believe that similar conclusions can be found in that case.

A more detailed discussion of disorder can be found in the revised manuscript, where we have included a new Sec.5, devoted to the role of disorder in the system.

... Further, the authors should comment more on potential realizations of their model in setups where the effects of disorder can be mitigated. These may include ultracold atoms or topoelectric circuits, which are mentioned only very briefly at the end of the paper.

...

2-Expand the discussion on meta-material realizations as potential ways of mitigating the effect of disorder

We agree with the Referee that our manuscript would benefit from an expanded discussion about metamaterials. Accordingly, we have added a new Sec. 6 in the revised manuscript that discusses topoelectrical circuits as well as ultracold atoms in more detail.

2-The authors introduce many abbreviations throughout the text, such as HSP, TRIM, RIM to refer to points in momentum space. The paper would be clearer if some of these are removed and the terms spelled out in full. 3-On page 7, I do not believe that the invariant characterizing the topological insulator should be called a mirror "Chern number." Chern numbers are integer and not Z2, and typically computed on even-dimensional (sub-)manifolds, whereas the authors consider 1D slices of the Brillouin zone. This terminology should be clarified in order to avoid confusion.

We thank the Referee for his suggestions and comments, which we have implemented in the revised manuscript.

We hope to have satisfactorily addressed the Referee’s concerns and suggestions and believe that our revised manuscript is now suitable for publication in SciPost.

---

## Round 2 · List of Changes

(All changes are marked in red in the main text)

• removed a few abbreviations (TPT, HSP, TRIM, RIM)
• replaced „mirror Chern number MZ2“ by „mirror topological invariant MZ2“ on p.7.
• added several new references with discussion (Kruthoff2017:PRX, Volpez2018:PRB,Volpez2019:PRL,Klinovaja2016:PRL)
• added linear conductance section with disorder
• added an additional discussion on topoelectrical curcuits (and other metamaterials)

---

## Editorial Decision

published